# Trauma Recovery Rubric: A Mixed-Method Analysis of Trauma Recovery Pathways in Four Countries

**DOI:** 10.3390/ijerph191610310

**Published:** 2022-08-19

**Authors:** Kleio Koutra, Courtney Burns, Laura Sinko, Sachiko Kita, Hülya Bilgin, Denise Saint Arnault

**Affiliations:** 1Social Work Department, Hellenic Mediterranean University, 71401 Crete, Greece; 2University of Michigan Medical School, Ann Arbor, MI 48109, USA; 3Department of Nursing, Temple University College of Public Health, Philadelphia, PA 19122, USA; 4Department of Family Nursing, Division of Health Science and Nursing, Graduate School of Medicine, The University of Tokyo, Tokyo 1130003, Japan; 5Mental Health and Psychiatric Nursing Department, Florence Nightingale Nursing Faculty, Istanbul University-Cerrahpasa, Istanbul 34381, Turkey; 6School of Nursing, University of Michigan, Ann Arbor, MI 48109, USA

**Keywords:** trauma, recovery, rubric, mixed method

## Abstract

Research is beginning to examine gender-based violence (GBV) survivors’ recovery, but little is known about diverse recovery trajectories or their relationships with other distress and recovery variables. This interdisciplinary, international multisite mixed-method study developed and used the TRR to identify and classify survivors’ trauma pathways. This study describes the phases of the initial development of the preliminary TRR (Phase 1), refines and calibrates the TRR (Phase 2), and then integrates the TRR into quantitative data from four countries (Phase 3). Seven recovery pathways with six domains emerged: normalizing, minimizing, consumed/trapped; shutdown or frozen, surviving, seeking and fighting for integration; finding integration/equanimity. Depression scores were related to most recovery domains, and TRR scores had large effect sizes. At the same time, PTSD was not statistically related to TRR scores, but TRR had a medium effect size. Our study found that the TRR can be implemented in diverse cultural settings and promises a reliable cross-cultural tool. The TRR is a survivor-centered, trauma-informed way to understand different survivorship pathways and how different pathways impact health outcomes. Overall, this rubric provides a foundation for future study on differences in survivor healing and the drivers of these differences. This tool can potentially improve survivor care delivery and our understanding of how to meet best the needs of the survivor populations we intend to serve.

## 1. Introduction

Gender-based violence (GBV) is violence perpetrated against another because of their gender and is held in social power systems. Women-identifying individuals are disproportionately impacted by GBV, including physical, sexual, verbal, emotional, economic, and psychological abuse in childhood or adulthood and threats or coercion based on a person’s biological sex or gender identity. It is estimated that one in three women worldwide experiences some GBV event in their lifetime. 

Recovery after GBV is rarely a linear process. Survivors use various methods to deal with the consequences of the trauma related to these experiences, often including diminished functioning, negative self-view, and lower quality of life. The consequences of GBV challenge survivors’ recovery long after the abusive relationship ended in many different life domains [1]. Specifically, the “lived GBV experience” can impact women physically, emotionally, and spiritually and change how women perceive themselves [2]. Changes in survivors’ self-view can influence their behavior and help-seeking actions, consequently impacting revictimization experiences or successful integration of the traumatic experience within their lives [3]. While healing is undoubtedly possible, capturing differences in the recovery pathways that survivors may experience as they navigate their healing journeys is often challenging. Thus, the purpose of this manuscript is to describe the development and testing of a rubric to capture pathways of trauma recovery after GBV.

## 2. Defining Recovery Domains and Criteria

Harvey criticized research assessing trauma recovery, noting that it has relied on poorly defined and seldom specified criteria [4]. Since then, progress has been made in defining the domains that can characterize successful recovery after traumatic events [2]. This paper defines recovery as regularly using skills, characteristics, or strengths that enhance health, security, and wellbeing. These skills or strengths include intentionality for the survivor to take action and attempt to “go on with normal life” [5], as well as seeking support from others to combat isolation and fulfill emotional needs [1]. Scientific literature also highlights the role of the informal support of family and friends in successful recovery from GBV [6]. For example, one survey indicated that decision-making about selecting sources of support is a vital recovery skill [7]. Supportive networks encourage survivors to increase their positive ties and set boundaries on toxic relationships to promote mental health and support recovery [8]. 

A qualitative meta-synthesis of survivors’ perspectives of GBV recovery found that trauma recovery domains are multidimensional, requiring courage, active engagement, and patience [9]. The five primary domains of the healing process are (1) trauma processing and reexamination, (2) managing negative states, (3) rebuilding the self, (4) connecting with others, and (5) regaining hope and power.

Explicit criteria for successful recovery are essential for assessment, treatment, and research. Despite naming the domains that should be evaluated when assessing recovery “models of trauma resolution”, Lebowitz et al. note that there are often insufficient descriptions of what defines recovery within those domains [10]. To define recovery criteria, Sinko and Saint Arnault [5] used qualitative methods to extract definitions of recovery for trauma survivors. They discovered three interconnecting recovery objectives: reconnection with the self, others, and the world [5]. Reconnection with the self involves reclaiming one’s identity and making decisions autonomously. Reconnection with others involves feeling a sense of belonging in the community. Reconnection with the world involves developing a positive view of the world and finding fulfillment and personal growth. In addition to these recovery criteria, a 2020 review of recovery after intimate partner violence, a form of GBV, described developmental aspects of recovery, which included disentangling from the past, coping with the present, and moving toward the future [6]. 

### 2.1. Measuring Trauma Recovery Using Domains and Criteria

Most trauma recovery measurement literature has used the absence of psychological symptoms such as depression, PTSD, and other clinical distress to indicate trauma recovery [11]. However, research is beginning to move away from measuring symptoms, service use, or clinician-based recovery assessment because they are based on medical models of mental illness, which may conflict with the survivor’s definition. This research conceptualizes trauma recovery as a process representing a movement toward integrating a healthy and thriving self [12]. For example, one survivor-oriented definition of psychological recovery is “establishing a fulfilling, meaningful life and a positive sense of identity founded on hopefulness and self-determination” [13]. Within this vein, Harvey [4] describes eight recovery domains, including Authority over remembering, Integration of memory and affect; Affect tolerance and regulation; Symptom mastery; Self-esteem; Self-cohesion; Safe attachment; Meaning-making [4]. She developed a clinician-rated trauma recovery measure rated on a scale ranging from relatively poor to quite good.

Yet, more recently, there has been a trend toward a more holistic approach incorporating positive recovery outcomes [14]. For example, one study found that successful trauma recovery involves the experience of “breaking free” [15]. Another study categorizes successful trauma recovery as “an upward trajectory” [16] and labels those who have recovered as “thrivers” [16]. From this, Wanner et al. [17] developed a 43-item trauma-specific quality of life measure that evaluates the five successful outcomes, including Emotional Well-Being, Functional Engagement, Recovery/Resilience, Peri-Traumatic Experience, and Physical Well-Being [17]. In addition, Tedeschi Blevins and Riffle have operationalized the concept of posttraumatic growth with domains of new possibilities, relating to others, personal strength, spiritual change, and appreciation of life [18]. For survivors of GBV specifically, Sinko, Schaitkin, and Saint Arnault [19] have introduced a Healing After Gender-based Violence instrument, which attempts to holistically capture healing as an outcome. However, these instruments do not capture the recovery pathways or explain relationships with other healing variables [19]. This study defines recovery domains and criteria by looking at the range of recovery, examining recovery not as an endpoint by pathways or phases, leading to desired recovery outcomes. 

Research that examines trauma recovery from a process (rather than outcome) point of view tends to reference “pathways” of trauma recovery [20]. Herman wrote, “recovery unfolds in three stages…the first stage is the establishment of safety…the second stage is remembrance and mourning, and the third stage is reconnection with ordinary life. Treatment must be appropriate to the patient’s stage of recovery” [20] (p. 99). Other research on mental illness recovery has taken the same approach, describing stages of mental illness recovery as a time of moratorium or withdrawal, awareness, preparation, rebuilding, and growth (characterized as living a full and meaningful life, self-management of the illness, resilience, and a positive sense of self) [13]. Another frequently used metaphor for trauma recovery stages includes stages of “integration” or “self-integration” [21]. This recovery model refers to the self-integration stage in which the survivor has regained possession or control of something stolen or lost [22]. This integration includes regaining the self and integrating the impact of the trauma as a part of that new self [21]. This ultimate stage of recovery as self-integration echoes other stages of recovery, such as empowerment [23], becoming resolute [16], and reconnection with the self [5,20]. While these stages have been theorized about, there is limited knowledge about holistically assessing the pathway of recovery. In addition, some stages mentioned, such as reconnecting with ordinary life in Judith Herman’s model, are complex processes that may require additional exploration to articulate variations and benchmarks within this pathway. These gaps in understanding call for building hypothesized stage or pathway models that can be used for assessment. 

### 2.2. Using Rubrics to Measure Trauma Recovery for Mixed-Method Research

Qualitative measurement of participants’ perception of their trauma recovery aims to understand a specific individual at a specific time [24]. The limited understanding of the trauma recovery processes of GBV survivors calls for more research, primarily focusing on identifying the domains and pathways of trauma recovery from the survivor’s perspective. Starting from the position of listening deeply to the experiences and perspectives of the survivor’s exploration of significant points in their journey, trauma recognition, speaking out, external motivations, and social support/pressure [1,9] can give researchers and clinicians a deeper understanding of their unique healing journey. GBV survivors’ realities, interpretations, experiences, and voices must be part of the egalitarian dialogue and not reinterpreted by experts [1,9]. This holistic inquiry focuses on what matters most to the patient rather than the narrow confines of the clinical checklist [25,26]. 

Moreover, this research combines participants’ perceptions of their recovery alongside quantitative measures of recovery with a mixed methods research design. To do this, we needed a way to quantify qualitative data [27]. Rubrics provide a framework for operationalizing the achievement of a process, and their use is emerging in health research [28] and provide a quantitative metric for use in mixed methods research. Rubrics quantify qualitative data by defining and explaining terms, clarifying process dimensions, codifying distinctions, and defining process achievement levels. 

The purpose of this study was to develop a Trauma Recovery Rubric (TRR) to quantify trauma recovery domains and pathways for a sample of GBV survivors and to examine the relationship between the TRR scores against quantitative measures of trauma recovery challenge indicators (PTSD and depression symptoms) and trauma recovery indicators (posttraumatic growth and sense of coherence). In this three-phase mixed-method study, we describe the initial development of the preliminary TRR (Phase 1), the refinement and calibration of the TRR (Phase 2), and the mixed-method integration of the TRR scores with quantitative data from GBV samples in four countries (USA, Japan, Greece, and Turkey) (Phase 3). Our Phase 3 hypotheses are (1) the TRR score will relate positively with recovery indicators and negatively with recovery challenges, (2) the TRR score will explain distress symptoms in a regression model, and (3) TRR scores will have a clinically significant effect on distress scores.

## 3. Materials and Methods

### 3.1. Design and Sample

This multisite mixed-method study is part of a more extensive international multisite project aimed at understanding the relationships between culture, help-seeking, and trauma recovery for survivors of GBV. All participants for all phases were recruited by investigators in their local sites using flyers in community agencies providing women with violence-related services and word-of-mouth. The sample for all phases was over 18, and they could read and write in their language. All participants in all phases of this study self-identified as women and self-identified as a survivor of some form of GBV. All qualitative data for all phases were collected using the same narrative interview of the Clinical Ethnographic Narrative Interview (CENI) protocol [29] (see instruments below). All phases were approved by the University of Michigan IRB (HUM HUM00144780) for the sexual assault survivors, and the Japanese site was approved by the University of Tokyo Graduate School of Medicine Ethical Committee (11756–2).

The Phase 1 sample consisted of datasets from two different survivor samples (N = 33). One sample was 19 sexual assault US survivors ages 18–26 [1]. Twelve women were current undergraduate students at a four-year institution, and seven were alumni. Thirteen identified as Caucasian, three women identified as African American or Black, and three identified as Asian. A total of 8 women had a history of childhood abuse, 16 had a history of unwanted sexual intercourse when enrolled as undergraduates, and 18 disclosed other forms of unwanted sexual contact as undergraduates. The second sample was 24 Japanese women sampled from domestic violence service centers in the Tokyo metropolitan area. They were 36–59 years of age, and most had children. All the participants reported domestic abuse (emotional, physical, or sexual). 

The Phase 2 sample used interview data from 24 US GBV survivors [5] and 12 Irish domestic abuse survivors [1] to refine the rubric and achieve interrater concordance. The US GBV survivors’ ages ranged from 20 to 81 years, and about one-half had children. Eighteen women identified as Caucasian, two women identified as African American, and one identified as Asian. Half of the sample had experienced more than one type of GBV; five had experienced child abuse, ten had experienced sexual abuse, and nine had experienced domestic abuse. The second dataset was from twelve Irish domestic abuse survivors. The Irish survivors were primarily from rural areas of Ireland, ranging from 20 to 64 years old, and all had children. Eleven women identified as Caucasian, and one identified as Black. All but one of the women was born in Ireland; however, several had lived abroad, primarily in England. All twelve participants reported abuse (emotional, physical, or sexual) in childhood and domestic abuse. 

The Phase 3 mixed-method sample consisted of 23 IPV survivors from Japan, 22 sexual assault survivors from the USA, 14 GBV survivors from Greece, and 12 from Turkey. The US and Japanese samples are described above. The thirteen Greek GBV survivors from Greece ranged from 22 to 60 years old. All participants were native Greeks; the majority had children, experienced domestic abuse, and had never sought help at some official health and social care service regarding the violence they were experiencing at the time. Most lived in an urban center; however, some spent significant time in rural areas in Greece. Most were employed, and education ranged from secondary education to postdoctoral training. The 12 Turkish survivors ranged in age from 25 to 40. Most lived in urban areas and had a college degree and children up to 5 years old. Half of the sample had experienced sexual, physical, or psychological-emotional violence. 

### 3.2. Measures

#### 3.2.1. Qualitative Instruments

Clinical Ethnographic Narrative Interview (CENI): We conducted semi-structured interviews following the CENI protocol [29]. The CENI uses experiential tasks to illuminate cultural, social, and personal meanings ascribed to the recovery process after GBV. The experiential tasks include creating a social network map, a body map, a lifeline, and a card sorting task. The structure of the CENI facilitates the production of narratives, and the interviewer’s role is to facilitate the process by offering deep listening without interpreting the material [29]. 

Trauma Recovery Rubric: The development and evaluation of the Trauma Recovery Rubric (TRR) is the focus of this paper. The final version of the TRR includes seven trauma recovery pathways: avoidance (normalization and minimizing), coping with memories and feelings (consumed, shutdown, and surviving), and regaining mastery and health (seeking integration and finding equanimity). Each recovery phase has criteria that characterize the six domains of trauma recovery: trauma definition; balancing emotions, body, cognition, and behavior; acceptance of trauma impact; holistic self-view; autonomous functioning; engagement with a supportive social network. The ratings included domains and sum scores (the scale and scoring are included in Appendix A). 

#### 3.2.2. Quantitative Measures

Recovery challenges were measured with the Physicians Health Questionnaire 8 (PHQ8) depression scale and the PTSD Checklist for the DSM5 (PCL–5) screener. The PHQ8 is an 8-item depression scale that does not include the suicidal thoughts item in the PHQ−9, making it suitable for survey use [30]. The eight-item version of the Patient Health Questionnaire (PHQ−8) [31] is a valid diagnostic tool that measures depressive symptoms in the general population. Respondents are asked to assess the frequency of symptoms in the past two weeks on a 4-point response scale from 0 (“Not at all”) to 3 (“Nearly every day”), resulting in a total score range from 0 to 24. A clinical cut-off score of ≥10 has been recommended to indicate probable depression. A score of 10 or higher on the PHQ8 indicates probable major depression. Cronbach’s alpha reliability of the PHQ8 in our studies ranges from 0.87 to 0.94 [32,33,34].

The PTSD Checklist for DSM−5 (PCL-5) [35] is a widely used and validated measure that assesses the presence and severity of PTSD symptoms. The measure consists of 20 items that correspond with DSM−5 criteria for PTSD. Respondents are asked to rate how bothered they have been by the symptoms in the past month on a 5-point response scale from 0 (“Not at all”) to 4 (“Extremely”), resulting in a total score range from 0 to 80. A clinical cut-off score of ≥31−33 has been recommended to indicate probable PTSD [35,36]. Our studies’ Cronbach’s alpha reliability of the PCL-5 ranges from 0.94 to 0.96 [32,33,34]. 

Recovery indicators include a sense of coherence and posttraumatic growth. The shortened version of the Orientation to Life Questionnaire (SOC-13) [37] is a widely used measure that assesses the sense of coherence, a concept at the heart of the salutogenic model of health and argued to be an important determinant of successful coping with stressful life situations [38,39]. SOC-13 consists of 13 items about how people view their life, measuring the three main components of a sense of coherence: comprehensibility, manageability, and meaningfulness. Participants rate their level of agreement or disagreement on a 7-point semantic differential scale, with two anchoring responses adjusted to each item. The total score range is from 13 to 91, and a higher score indicates a stronger sense of coherence. Cronbach’s alpha for SOC-13 in our sample was 0.82–85 [32,33,34]. 

Posttraumatic growth was measured by the Posttraumatic Growth Inventory (PTGI), a 21-item self-report instrument used to assess psychological growth following a traumatic event [18], a 21-item self-report instrument used for assessing psychological growth following a traumatic event. The PTGI has five subscales: New Possibilities (e.g., “Established a new path for my life”); Relating to Others (e.g., “A sense of closeness with others”); Personal Strength (e.g., “Knowing I can handle difficulties”); Spiritual Change (e.g., “I have a stronger religious faith”); Appreciation for Life (“Appreciating each day”). Participants in this study were asked to indicate for each of the statements the degree to which this change occurred in their life since their most distressing or traumatic unwanted sexual experience as an undergraduate. The PTGI scores range from 1 to 126, with higher scores reflecting greater perceived growth. Items on the PTGI range from 1 (“I did not experience this change as a result of my crisis”) to 6 (“I experienced this change to a very great degree as a result of my crisis”). Cronbach’s alpha for our sample was 0.90 [33].

### 3.3. Data Analysis

#### 3.3.1. Phase 1: Instrument Development 

The development of this rubric came from the analysis of survivor trauma recovery processes in samples from two very different research sites (19 sexual assault survivors in the US and 24 domestic violence survivors in Japan). The senior author convened these researchers together (LS and SK) as an expert panel. They developed a rudimentary draft of “trauma recovery pathways”. The investigators carried out an analysis to capture the critical recovery domains and patterns. Once the qualitative coding themes were generated, an intensity matrix was developed to document the occurrence of themes [40]. A rating matrix was created using pathways across the top (giving names to the themes) and domains down the side (defining the theme’s criteria). 

#### 3.3.2. Phase 2: Instrument Refinement and Interrater Concordance

The next stage in the rubric development was refined, and interrater concordance was achieved. In this study, investigators coded the same survivors’ qualitative narratives and achieved concordance with their ratings. Two independent investigators (KK and CB) from the USA and Greece conducted this analysis. The qualitative datasets consisted of 36 interviews following the Clinical Ethnographic Narrative Interview (CENI) protocol [29]. To rate any given interview, the investigators rated each domain and totaled overall scores for the survivor. We refined the rubric definitions from each round. We used the domain and sum scores for our intercoder concordance. 

This process continued for five rounds until we achieved concordance of sum scores for four interviews. To arrive at our concordance metric, we were persuaded by McDonald and colleagues [41]. They wrote, “Quantitative researchers have sometimes made the mistake of evaluating qualitative research reports using the standards of quantitative research, expecting (interrater reliability) IRR regardless of the nature of the qualitative research. As a result, reporting statistical measures may be alluring for qualitative researchers who believe that reviewers unfamiliar with their methods will respond to IRR as a signal of reliability; however, for many methods, reliability measures and IRR do not make sense” [41] (p. 72:5). We continued our analysis and refinement until we achieved 100% interrater concordance of the sum scores. 

#### 3.3.3. Phase 3: Mixed-Method Integration

Next, investigators from four countries (Japan, US, Greece, and Turkey) applied the TRR to their countries’ CENI narrative interview database. Investigators worked with the data collected in their native language. The researcher was instructed to rate each participant for each domain and captured this data in online Qualtrics. The Qualtrics form also captured the raters’ comments about the holistic evaluation of the survivor’s recovery, the cultural dimensions they noticed as they applied the rubric to survivors from their culture, and difficulties or commentary about the clarity and usefulness of the TRR tool. The senior author merged the TRR database with the rest of the quantitative data for each participant, calculating the sum of TRR scores for each participant. Quantitative analysis was then carried out using the TRR ratings, recovery challenge indicators (PTSD and depression), and recovery indicators (posttraumatic growth and sense of coherence). Overall scores of all quantitative instruments were summed, and survivors were categorized into the trauma recovery pathway represented in their interview. 

We used correlational analysis to determine interrelationships among the variables. We used ANOVA to determine whether there were country-level differences in depression or PTSD or TRR scores. We also used *t*-tests to examine whether the mean scores differed between survivors who met probable depression or PTSD criteria compared with those who did not. Regression analysis was used to examine the relationship with overall TRR scores, recovery indices (sense of coherence and posttraumatic growth), and recovery challenges (PTSD and depression). Finally, effect size calculation was used to determine the effect size of the TRR score on the depression or PTSD scores for the entire sample [42].

## 4. Results

### 4.1. Phase 1: Instrument Development

Independent analysis of initial data revealed that both the American and the Japanese trauma survivors ranged in their achievement of the state of trauma integration. We analyzed the qualitative data to discover “how” the survivor incorporated their trauma into their lives, defined as “trauma integration”. During this analysis, we identified subgroups, including those who minimized the impact of the trauma on their lives, those who identified themselves “primarily” as a survivor, those who described feeling withdrawn, cut-off, an outsider in their lives, or emotionally and physically numb. Another subgroup described being unable to manage their symptoms and life events, seemingly in perpetual states of chaos or frustration. 

The final step was defining the domains the survivor needed to “integrate” their traumatic experience(s). These domains included the definition of trauma, acceptance of trauma, their ability to manage or balance symptoms, understanding the interactions among emotions, needs, situations, and others, feeling empowered, and a holistic self-view. Ultimately, our preliminary rubric comprised five pathways and five domains. 

### 4.2. Phase 2: Instrument Refinement and Calibration

Two researchers refined the initial rubric by applying the rubric independently to the same survivor’s narrative interview transcripts. A total of five rounds were performed, and after each round, changes were made to the rubric for clarification, and the new rubric was applied to new participants in subsequent rounds. Transcripts of participant interviews were selected randomly, and the investigators uploaded their coding blinded from each other. Using the final rubric, the team achieved 100% concordance of several pairs of sum scores. The changes in the rubric are shown in Table 1, and the final rubric is shown in Appendix A.

### 4.3. Phase 3: Mixed-Method Integration

#### 4.3.1. Recovery Pathways Rating and Description

First, each county investigator rated each survivor according to the rubric. Table 2 shows the analyst’s comments about the case, and some sample quotes from the interviews that substantiated the scores they gave and helps us see the differences among the groups. We found that the TRR rating was applicable to the survivors across all the countries. As seen in Table 2, survivors in the Avoidance pathways (normalizing and minimizing) resist self-awareness about the significance of or impact of their trauma in their lives or seeking help for it. Moreover, there were relatively few survivors in this category, probably because people in these categories tend not to enter studies such as ours. People in the difficulty managing pathways (consumed, surviving, and shutdown) showed that there were varieties in difficulty managing their trauma and that these difficulties did generally fall into three categories. The final group of pathways, the finding integration group (seeking and equanimity) showed significant resolution of their trauma, showing differences in the consistency of this integration. 

#### 4.3.2. Statistical Significance

Correlational analysis revealed no significant correlations between age and any study variable. There were significant positive correlations between PHQ8 and PCL5 (r = 0.68, *p* = 0.00). There were significant negative correlations between SOC and PHQ8 (r = 0.66, *p* = 0.00) and PCL5 (r = 0.66, *p* = 0.00). ANOVA comparison of means indicates country-level differences in PTSD scores with Turkey having a significantly higher PTSD score mean (F = 4.9 (df = 3), *p* = 0.00) (Table 3).

ANOVA comparison of means revealed no country-level differences in trauma integration scores (Table 2). PHQ8 was negatively associated with TRR sum (r = −0.29, *p* = 0.01), and all of the domains: trauma definition (r= −2.9, *p* = 0.02), trauma acceptance (r= −0.24, *p* = 0.04), balance (r= −0.27, *p* = 0.02), holistic self-view (r= −0.28, *p* = 0.02), empowered functioning (r= −0.31, *p* = 0.01), and social engagement (r= −0.24, *p* = 0.05). PHQ9 functioning mean was also negatively associated with TRR sum (r = −0.38, *p* = 0.01) and most of the domains: trauma definition (r= −32, *p* = 0.02), trauma acceptance (r= −0.39, *p* = 0.00), Balance (r= −0.38, *p* = 0.01), and empowered functioning (r= −0.43, *p* = 0.00). There was a significant positive correlation between TRR sum and SOC (r = 0.27, *p* = 0.02). SOC was also positively associated with trauma definition (r = 2.7, *p* = 0.02), trauma acceptance (r = 0.27, *p* = 0.02), balance (r = 0.27, *p* = 0.02), and empowered functioning (r = 0.31, *p* = 0.01).

Independent sample *t*-tests were used to examine whether the mean TRR scores differed for survivors who met the probable depression or PTSD criteria compared with those who did not. *T*-tests were significant for depression subgroups, with non-depressed survivors (n = 42) having significantly higher TRR means (M = 35.0, SD = 7.7) compared with their depressed counterparts (n = 29) (M = 29.6, SD = 8.7) (t = 2.8, df69, *p* = 0.00). However, there were no differences between probable PTSD and those without probable PTSD mean scores (Table 4). Figure 1 shows each recovery group’s mean depression, PTSD, and SOC scores.

#### 4.3.3. Prediction

Regression analysis shows that the TRR sum score predicted 8.7% of the depression score (t5.5, *p* = 0.01). Regression analysis also shows that the SOC score predicted 7.3% of the variance of the TRR sum score (t5.5, *p* = 0.02). Regressions for PTSD scores or PTG scores were not significant.

#### 4.3.4. Clinical Significance

We calculated the effect size for TRR by depression and PTSD groups. The Cohen’s d effect size of the TRR score was 0.37 for PTSD and 0.67 for depression.

## 5. Discussion

This mixed-method study had multiple goals to offer further insight into how we might capture variations in GBV recovery pathways and how these pathways influence women survivors’ abilities to reach their recovery goals. The rubric we created, the TRR, constitutes an attempt at determining, creating, and assessing pathways and domains that promote recovery engagement to further the study and analyze the GBV phenomenon. They used an innovative approach of applying the resulting rubric with a sample of narrative interviews from four countries to evaluate its use within different cultural environments. While our quantitative analyses revealed no country-level differences in trauma integration scores, we found differences when comparing survivors with clinically relevant depression with those who did not. We also found that depression and an individual’s sense of coherence significantly predicted one’s TRR score, but PTSD, in contrast, did not. This finding suggests that depression and PTSD have differential impacts on trauma recovery and warrants additional study. This rubric can be used to further understand recovery pathways cross-culturally. It can also allow researchers to examine differing recovery trajectories and other risk or protective variables.

The need for an instrument to capture trauma recovery pathways arose through the collaboration and discussion among the twelve countries within the larger international research consortium of MiStory (see https://mistory-traumarecovery.org/home, accessed on 14 May 2021). The TRR was created to analyze and quantify survivor narrative data using a rubric based on these discussions. To date, rubric scoring tools have mainly been used in the education sector to implement and evaluate specific assignments or tasks [43]. This study is the first to use the rubric for quantifying qualitative data in assessing trauma recovery. As such, this research could constitute a model for analyzing other similar research efforts.

Initially, our research group articulated pathways based on coded concepts within survivor narrative data that seemed to articulate different pathways of healing. The meaning women described for that healing and how they organized their recovery process led our researchers and raters to set criteria for women’s recovery, yielding seven pathways with six domains. Articulating the domains is related to defining broader healing experiences, and a rubric is a new development in the field. These domains are supported in part by previous literature. For example, the concept of internalized normalization, which contributed to our understanding of the “normalizing” group, has been articulated regarding specific GBV populations [44]. Due to often narrow legal definitions of GBV-related crimes and other social circumstances surrounding GBV, many survivors minimize their experiences or mistakenly believe that their experience is not severe enough to be reported [45]. Less is known about what characterizes the consumed, shutdown, or surviving recovery pathways, which is why we made this rubric. The subtle differences in “fighting for integration” and “finding equanimity” were interesting to clarify, and hearing survivors talk about finding peace has helped us gain a deeper understanding. Regardless of the specific pathways, readers are urged not to regard them as “destinations” but rather as pathways along a multidimensional journey [2,9,12].

Implementation of the final TRR version within four different cultural settings demonstrates that this tool has promise as a reliable cross-cultural tool. At the same time, it highlights some potentially universal aspects of gender-based violence recovery and the global consequences of violence against women on personal, social, and spiritual levels, impacting their ability to connect to themselves, others, and the world around them [5,44]. In fact, during the implementation of TRR in four different countries, the raters did not record problems in understanding or meaning, highlighting the global nature of the recovery themes. However, it should be mentioned that the way GBV recovery is manifested in each culture is often different. These cultural nuances represent important healing objectives that may influence these pathways [44]. Additionally, these pathways may be different in different forms of GBV trauma. In general, we advise caution when interpreting these results without understanding the cultural context [19,45]. Future research will use qualitative and mixed methods to discern the cultural differences and similarities. We anticipate that the differences will be more a matter of emphasis and meaning rather than in the pathways themselves, but this is unknown. Moreover, previous research has shown that issues such as normalization and shame seem to exist across cultures.

Regarding distress scores by country, we found that the PTSD scores for the survivors in the Turkish sample were high compared to the other county’s samples. We are investigating this finding further. However, research on domestic violence in Turkey has found very high rates of distress. For example, a study of 59 survivors living in domestic violence shelters in Istanbul found that 76.3% were diagnosed with at least one psychiatric disorder, and posttraumatic stress disorder was the most common diagnosis (50.8%). In addition, over half of the women in their sample had attempted suicide at least once, and 66% of these were found to have attempted suicide after the violence started [46]. Much more research that systematically compares levels of distress in GBV survivors is needed.

Our scoring of the TRR application in the 71 CENI interviews revealed a lack of normalizing and minimizing pathways in all four countries. This finding may have to do with the likelihood that only those who acknowledge the impact of trauma would volunteer for GBV research [19]. Likewise, the large number of “integrated” survivors might be because the peace and security in their recovery allow them to volunteer for research as part of “giving back” to the community [19].

Quantifying qualitative data using the TRR allowed us to integrate other quantitative data, including depression, PTSD, sense of coherence, and posttraumatic growth. Our hypothesis that TRR would be negatively related to recovery challenges of depression and PTSD was confirmed. This finding is consistent with data reported by others that GBV survivors are regularly reported to have higher PTSD and depression [47]. This result shows the promise that the TRR is capturing what it should. Moreover, we hypothesized that the TRR scores would explain some variations in the trauma recovery challenge indicators, and this hypothesis was partially upheld. Finally, we hypothesized that the TRR score would have clinically significant effect sizes and found moderate effect sizes for PTSD and large effect sizes for depression. This finding also suggests the tool’s clinical significance. While it is known that being exposed to GBV is predictive of depression [48], less is known about how depression is interrelated with factors such as low self-esteem, less social contact [20,49], or emotional self-regulation.

Our hypothesis that positive associations would be between TRR scores, PTG, and SOC was partially supported. SOC includes meaning and purpose in life and the general orientation toward “understanding” one’s life, which overlap with the integration pathways of the TRR. However, while the PTG description references a sense of self-understanding, we did not find an association between TRR and PTG. More research is needed to disentangle the overlap between trauma integration, other health indicators such as wellbeing, sense of coherence, and posttraumatic growth after GBV and other traumas.

We believe that this study illustrates that practitioners can use qualitative narrative data to understand the recovery pathways of their clients. Narrative methods allow practitioners to listen to the experience of survivors from a holistic view, hearing what matters most to the client, and can then be followed by other assessment tools and clinical checklists. Our rubric used qualitative data to operationalize the dimensions of recovery, allowing us to compare subgroups of survivors. Using narrative methods, rubrics, and standardized measures to identify subgroups may help refine clinical interventions tailored to survivors’ needs.

## 6. Conclusions

While we used narrative data from multiple countries, generalizability to countries whose narratives were not incorporated in this analysis is limited and warrants further testing. Additionally, this study was specific to adult-aged women-identified survivors, and caution should be used before applying these concepts to children, men, or non-binary survivors. The study’s cross-sectional design precludes any causal relations, and we were limited by the variables gathered in our initial dataset. Understanding how other recovery-oriented measures correlate with these domains may give insight into what drives healing variation in survivor groups. Future research should consider using this rubric longitudinally with larger samples to understand change over time.

In this study, the raters intentionally did not follow clinical or other diagnostic criteria when analyzing survivor narratives to avoid the potential for diagnostic attempts to affect the pathway ratings. This strategy significantly differs from other studies looking at recovery benchmarks [4,15]. We vigorously resist the idea that survivors who score as more integrated are more “healed”. On the contrary, we expect the TRR to help tailor intervention approaches in clinical and support settings according to areas of concern at the scoring time. The TRR might also be used to expand research related to healing after GBV, like the emerging research on trauma recovery [2,5,6].

In closing, this study demonstrated that the TRR has significant promise as an innovation for analyzing and evaluating women’s GBV trauma recovery pathways and is a survivor-centered, trauma-informed way to understand and quantify pathways of recovery using narrative data. We hope the TRR inspires scientists to recognize and further investigate variations in survivor healing, to move away from a “one size fits all” approach. Another significant strength of TRR is that it combines narrative data from multiple countries, thus highlighting its cultural adequacy when used in differing cultural contexts. The TRR can be used to understand survivor narrative data, different survivorship pathways, and how different pathways may impact health outcomes. The TRR can be used clinically to understand better survivors lived experiences and where they are in their recovery processes at the time of scoring, revealing to survivors a direction or set of goals or an area to focus on in treatment, and at the same time, helping clinicians better tailor their approach.

## Figures and Tables

**Figure 1 ijerph-19-10310-f001:**
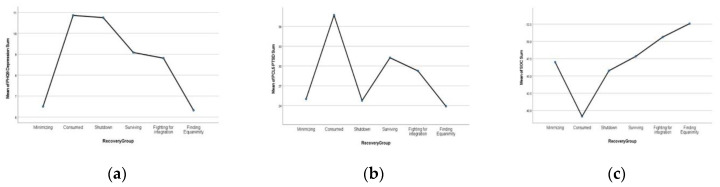
Recovery group’s means: (**a**) depression; (**b**) PTSD; (**c**) SOC scores.

**Table 1 ijerph-19-10310-t001:** Summary of rubric refinements.

Analysis Round	Major Discovery	Change Made to the Rubric
1 and 2	Definitions of “consumed” and “surviving/overwhelmed” were similar	The consumed pathway survivors struggled with the frustration of recovery; surviving pathway struggled with frustrations with navigating the challenges of daily life.
	Descriptions of “cut off/frozen” and “minimizing” were similar	Minimizing pathway trauma history was not relevant in the lives; cut off/frozen pathway sense of feeling stuck.
3	The philosophy and purpose of the rubric itself were unclear	Focus is on trauma only, not all of life’s adversities; assess how the survivor understands trauma in the fabric of her life and selfhood.
	Need to “bracket” psychiatric labels	Focus is on the survivors’ perceptions, not a diagnosis; removed any pathologizing phrases.
4	Social network engagement embedded in other domains	Made a new domain called “engaging with a supportive social network” which includes trust and mistrust, self-disclosure, social support, and interpersonal boundaries.
5	Needed to distinguish between “struggling to become integrated” and achieving “integration”	Service use, social support, or managing symptoms are necessary steps toward integration, but integration includes self-acceptance, a feeling of peace, and equanimity.

**Table 2 ijerph-19-10310-t002:** Quotes from the narrative rating analysis.

Main Categories	Pathways Quotes	Pathways Description
**Avoidance**	**Normalizing** (There are very few of these people in our dataset. We believe that is because they may believe that, since abuse is normal, they do not self-identify with the experiences, and therefore do not enroll in our studies.)	
	She seems to engage in self-blame and is desensitized to repeated unwanted sexual experiences, saying “I am kind of used to it… I see trends happening, and I see how boys think… I should have been more careful.” She still feels shame, embarrassment, and resentment, but also has accepted/resigned to her experiences. She feels helpless and abandoned, saying that she can’t talk to her friends about her unwanted experiences because “if you tell your friends, they are going to be like ‘if this is a pattern that you see, why don’t you just stop?’” She talks about feeling sad, blue, gloomy, distressed, and more, and expresses “I don’t know how to feel, and I don’t know the appropriate response as to what happened.” She hasn’t really engaged in any formal healing methods, saying that her primary method is to try to “brush it off”, but that she generally isn’t sure where she should go from here, and that she “isn’t really one to talk about her feelings”.	These people do not believe that the survivorship concept applies to them because what happened is normal. However, they sometimes focus on the social fallout that can occur if you were to name it and believe the will (or even should) be blamed.
	**Minimizing** (The comment for the normalizing pathway is also somewhat true of the people on the minimizing path. However, for them, it is not so much that it did not happen or is normal, but rather that talking about it is unproductive, too painful, or unnecessary.)	
	She says she isn’t close enough with any of her friends to talk with them about her feelings if she is having a hard time, and she states that she struggles to identify any causes of the strong emotions that she feels. She is very disconnected from her abuse history, saying, “It was right before I left for college. I blot it out of my memory, so I didn’t even think to write it on [lifeline], because I completely push it out.” “I have been totally ignoring it, and I almost don’t even want to open that box, because if I do then it will become real, and I will actually have to deal with it.”	These survivors want to push the trauma out of their minds and lives and describe being unable to deal with it.
	She strongly refused to see and talk about the impacts of trauma on her life, functioning, and physical and psychological health, although she knew the severe trauma happened in her life. In addition, she drew so many social networks around her; however, nobody was truly connected with her (very superficial and she did not talk about her suffering with anybody including her family). She tried to keep functioning in daily life by denying and rejecting to face the trauma.	
**Difficulty Managing**	**Consumed**	
	She has a very long history of assault and abuse going all the way back to high school. She elaborates so extensively about these continued experiences that she seems very consumed by them, and it has seeped into her professional life as well. She says that she still feels abandoned, helpless, hostile, anxious, and angry. She also says being a survivor is “a really big part” of her identity.	These survivors tend to describe survivorship as central to their identity and cannot see life without symptoms and struggles to heal.
	She is really, really struggling. She describes herself as being “very consumed by her feelings from her trauma.” She feels she has not progressed very far along the healing journey and feels like she is a failure like she has “amounted to nothing”. She cannot see a future where she has healed.	
	Trauma is the center of her identity. She seemed to live in trauma and be controlled by trauma effects. She always recognized her all physical and psychological distress, or a little stress in daily life was caused by her trauma in past. Although she tried to seek help from many people and connect with them, she believes nobody understands her.	
	She seemed to think she is different from other people, and nobody can understand her. She was always focusing on her trauma and its impacts. She seemed to be living with her trauma in a very deep hole inside her that nobody can’t reach.	
	**Shutdown**	
	She does well in school and goes to parties and things on the weekends but feels very disconnected from her social network. She has had a series of unwanted sexual experiences that she feels like are in a gray area of being unwanted, but that they made her feel uncomfortable. She feels caught in a pattern of dating/hooking up with guys and is having trouble breaking out of a pattern that she knows is clearly harmful to her. She talks about sadness, loneliness, despair, and feeling like she can’t move on.	These survivors tend to focus on feeling paralysis, being stuck, and feeling disconnected from themselves and others.
	She hasn’t talked about her trauma before this interview and says that it affects her life now in terms of hypervigilance, and that she isn’t very open with the people in her life. She says, “I still want to vomit sometimes when I think about it”, but seems very disconnected and numb from the experience, even saying “I just don’t care.”	
	This participant is struggling a lot and “I have been feeling broken”). She feels disconnected from others and is struggling with whether or not to seek help. She says, “It feels like nobody is there for me” and “I am not important in everybody else’s life, for them to ask me what is going on… I feel withdrawn… abandoned…” She also says that she is “scared that this is how [life] is going to be… immobilized…”	
	**Surviving**	
	She is focused on her economic problems and the problems in her and her family’s life. She does not connect these with traumatic events or trauma effects.	These survivors are generally exhausted, angry, feel betrayed by others, and are frustrated in their daily lives. They tend to have overwhelming problems, which are sometimes symptoms but usually life experiences.
	She feels a lot of anxiety and depression on a regular basis and holds a lot of tension in her body and exhibits a lot of physical symptoms. She has seen various social workers, therapists, and psychiatrists, but still experiences anxiety/depression daily. After the violence she experienced, she was not supported by the few people that she told.	
	She was very struggling and surviving in several and serious problems among her two children, such as violence against her and suicide attempts as the impact of trauma on them. She was also struggling from several physical and emotional distress because of being exhausted by the present problems. Although she had such significant social, physical and psychological problems, she try to cope the problems by connecting with trustful others and believe a hope for resolving the problems and getting her life better in the future.	
	She felt very overwhelmed and unable to control her symptoms in her daily life and health, including feelings of dissociation. She described struggling and surviving day-by-day. She did not try to have a close relationship with anybody because of a fear that she did not want to be hurt anymore and the belief that nobody understands her.	
	She felt strong frustrations, anger, and irritation with isolation and prejudice against a single mother and victim of IPV in the community she was living. She did not have self-confidence and strongly wants to be accepted by other people. She expressed she was surviving in such situations and had several emotional and body distress at present.	
**Achieving integration**	**Seeking**	
	She has great self-esteem, is well organised, and feels positive for her future. Still, she questions why she feels so vulnerable. She feels that it is ok to be vulnerable but wants to get over it and move on.	These survivors have significant improvements in many aspects of their lives but are still struggling with symptoms or difficulties. They emphasize a degree or mastery which gives them hope that they will achieve the healing they seek.
	She has a rich social network and a very healthy relationship with her fiancé. However, she experiences a lot of somatic symptoms, including pain. She describes her mental status very eloquently, stating, “I feel like I constantly have a storm in my mind. You know, sometimes it is just cloudy overcast, and sometimes it is a thunderstorm of both worry and stress, and just kind of anxiety overall. She is still feeling “withdrawn from who she is”, and a lot of spiraling anxiety, fatigue, and more. However, she acknowledges that this is a significant improvement from what she used to be like, saying that these negative feelings don’t overwhelm her to the degree that they used to.	
	I found this participant to be very well-spoken and reflective about her experience in an abusive relationship during college. She recognizes some long-lasting effects of her abusive relationship, saying that it somewhat affected all of her subsequent relationships with a heightened sense of vulnerability. While she has moved on, she admits that she ‘still kind of struggles sometimes’ with feelings of guilt, shame, and anger, but is doing very well overall.	
	Although she was struggling with several physical symptoms due to her trauma, she tried to understand why she has such symptoms and the connection with her trauma. In addition, she really appreciated for people around her who helped her out, such as her child, parents, peers, and friends, and could feel a deep connection between herself and others. She also felt the trauma was her past thing, and she want to move forward living with trauma.	
	She was beginning to recognize the importance of paying her attention to her emotions and body. She was still fragile and felt fear to connect with others, especially others who threatened her such as her mother who always dominated her, however, she was building up a sense of trust and hope with others throughout having a relationship with peers.	
	She still felt a difficulty of developing boundaries in the relationship with persons who threaten her and did not feel connections and integration between body, emotions, bad parts and good parts in her life, and trauma well. However, she was exploring to integrate those and trying to cope the difficulty in her relationship with others. She has hope that she can finally develop the ability to manage and cope and have a better life.	
	**Equanimity**	
	Her relationship with the people in her social network “is on a new level” and very positive because of her sharing with them about her experiences. She also exercises and engages in arts and crafts and is successfully studying mental health counseling. “There is a Hemingway quote that I really like. ‘The world breaks us all but some of us are stronger in the broken places’”. She reflects on healing saying it involves “realizing that I probably will get hurt again, probably in different ways, but that I can come back from it, that I am resilient.”	These survivors have a sense of calm, positivity, and security in their healing. They tend to emphasize their connection with themselves as well as with others.
	She has healthy coping strategies (yoga, cooking, talking with friends) and has a healthy relationship with her boyfriend. Her friends were also very supportive of her after these situations, and she said that helped her get over feelings of shame and self-blame for being in those situations in the first place.	
	She seemed very calm and easy during talking about her story and present condition. She strongly felt that her children and co-workers helped her out from the very dark phase of her life and deeply appreciated for them. She did not report serious physical and emotional symptoms and she said she can believe herself who could overcome such struggling phase and trauma.	
	She understood and could explain the connections between her emotion, body, and trauma. In addition, she felt the deeply connected with trusted others, such as her children, parents, and peers and those people healed herself with trauma. She expressed the trauma as a traffic accident in the past (indicating trauma was outside herself at present). She seemed very calm, positive, and active. She also said she loves herself and people surrounding her.	

**Table 3 ijerph-19-10310-t003:** Sample characteristics GBV survivors (N = 71) and pathway of trauma recovery per country.

Variable	Japan (N = 23)	Turkey (N = 12)	Greece (N = 14)	USA (N = 22)	
	M (SD)				
Age	46.5 (6.3)	31.7 (5.9)	42.5 (14.4)	22.1 (2.3)	
PHQ8	9.1 (7.3)	8.8 (4.3)	8.1 (5.3)	7.3 (5.8)	
PCL5	28.0 (20.3)	44.3 (13.3)	22.2 (13.8)	22.2 (17.2)	
SOC	43.8 (14.7)	49.9 (9.7)	53.6 (13.2)	52.1 (14.9)	
Current medical diagnosis	81%	66%	50%	66%	
**Trauma Recovery Pathway by Country Group**					**Total**
Normalizing	---	---	---	---	
Minimizing	---	---	---	---	
Consumed	4	---	1	2	9.9%
Shutdown	---	1	3	4	11.3%
Surviving	2	4	5	1	16.9%
Fighting for integration	6	2	2	6	22.5%
Finding equanimity	11	5	3	9	39.4%

**Table 4 ijerph-19-10310-t004:** Independent sample *t*-tests of mean TRR sum and domain score by depression and PTSD subgroup.

**Domain**	**Depression (PHQ8)**			**PTSD (PCL5)**		
	**Not Depressed** **(N = 42)**	**Probable Depression** **(N = 29)**	** *p* **	**No PTSD** **(N = 43)**	**Probable PTSD** **(N = 28)**	** *p* **
TRR sum scores	35.0 (7.7)	29.6 (8.7)	0.00	----	----	----
Trauma definition	5.9 (1.4)	4.8 (1.6)	0.01			
Balancing emotions, body, cognition, and behavior	5.8 (1.4)	4.9 (1.5)	0.02	----	----	----
Acceptance of trauma impact	5.7 (1.5)	4.8 (1.6)	0.03	----	----	----
Holistic self-view	6.0 (1.3)	4.9 (1.5)	0.00	----	----	----
Autonomous, empowered functioning	5.9 (1.3)	5.1 (1.5)	0.01	----	----	----
Engagement in a supportive social network	5.8 (1.6)	5.0 (1.6)	0.04	----	----	----

## Data Availability

Data are available upon request from the senior author.

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
