# Peer review of "Trauma Recovery Rubric: A Mixed-Method Analysis of Trauma Recovery Pathways in Four Countries"

_ijerph, 2022, doi:10.3390/ijerph191610310_

Round 1

Reviewer 1 Report

This is a very well written innovative article and very much needed in motivating a holistic approach to mental health following trauma from a variety of causes. Unfortunately , I have not been able to change my computer setting to send comments to specific areas anonymously .

However, it would be useful to reexamine some terminology , for example , as usual we are referring to stages in response to trauma as if there are in fact distinct and one simply moves from one stage to the other. Maybe though the path is really much more complex, i.e, there are up-and downs and back and forth! Not a static model .

In addition , although I concern with not getting stuck in medical diagnosis, we it be useful to recognize that in addition to cultural factor , there are certain known parameters that are part of the "persona" like resilience - in a biological sense. This is a well demonstrated phenomenon which should be somehow consider in the rubric approach.

Author Response

Dear Reviewer, thank you so much for your comments. We have revised the text and highlighted the main changes in red. Below we have responded to individual comments. 

Reviewer 2 Report

Dear Editors:

Thank you for sharing with me this manuscript, which reports on a novel approach (conceptually and methodologically) to assess trauma recovery, capitalizing on the very rich narratives that individuals who have suffered gender-based violence share. I also commend the authors for condensing so much work into a relatively short article; achieving this requires considerable effort.

My few recommendations that might help further strengthen the manuscript follow:

1-The rubric as presented in Table S2 is comprehensive, yet somewhat difficult to understand for anyone who has not implemented the kind of approach the authors are describing. It is not entirely clear how stages and domains are captured/assessed and how the researcher/clinician/social worker would go about using this rubric. Annotating the rubric to indicate how the person using it progresses through it (and explaining the rationale) would be helpful.

2-I am fascinated by the emphasis on narratives in this approach, but was very disappointed, surprised even, that there are no narratives shared in this paper, making it difficult to appreciate how narratives inform the work described. While the authors underscore the value of such a qualitative approach, the report ends up giving priority to quantitative data. (Also, the authors describe a mixed-methods approach, yet could provide a clearer rationale for "mixing" methods; see, e.g., Creswell and Plano-Clark, 2017).

3-On a related note, by de-emphasizing the qualitative data, the authors have given up on a opportunity to complete a more in-depth assessment of the role of context, including the cultural context, which is treated superficially. At the very least, I think it would be helpful to acknowledge this as a significant limitation of the work reported on here and something worth pursuing in the near future.

4-Finally, although there are references to how researchers obtained informed consent and how the study was reviewed by research ethics committees in the case of two of the sites, there is no complementary information about the others. Apologies if I somehow missed it, but this would be useful information to provide, especially as this work is intended for an international audience and, of course, given the vulnerable nature of the population with which the new tool is meant to be used. 

Author Response

(The authors gave the same response as above.)
